# A Combined Metabolome and Transcriptome Reveals the Lignin Metabolic Pathway during the Developmental Stages of Peel Coloration in the ‘Xinyu’ Pear

**DOI:** 10.3390/ijms25137481

**Published:** 2024-07-08

**Authors:** Cuicui Jiang, Keliang Lyu, Shaomin Zeng, Xiao’an Wang, Xiaoming Chen

**Affiliations:** Fruit Research Institute, Fujian Academy of Agricultural Sciences, Fuzhou 350013, China; lvkelianglkl@outlook.com (K.L.); zengshaomin@faas.cn (S.Z.); wangxiaoan@faas.cn (X.W.); chenxiaoming@faas.cn (X.C.)

**Keywords:** *Pyrus pyrifolia*, peel color, metabolome, transcriptome, lignin biosynthesis, co-expression analysis

## Abstract

Sand pear is the main cultivated pear species in China, and brown peel is a unique feature of sand pear. The formation of brown peel is related to the activity of the cork layer, of which lignin is an important component. The formation of brown peel is intimately associated with the biosynthesis and accumulation of lignin; however, the regulatory mechanism of lignin biosynthesis in pear peel remains unclear. In this study, we used a newly bred sand pear cultivar ‘Xinyu’ as the material to investigate the biosynthesis and accumulation of lignin at nine developmental stages using metabolomic and transcriptomic methods. Our results showed that the 30 days after flowering (DAF) to 50DAF were the key periods of lignin accumulation according to data analysis from the assays of lignin measurement, scanning electron microscope (SEM) observation, metabolomics, and transcriptomics. Through weighted gene co-expression network analysis (WGCNA), positively correlated modules with lignin were identified. A total of nine difference lignin components were identified and 148 differentially expressed genes (DEGs), including 10 structural genes (PAL1, C4H, two 4CL genes, HCT, CSE, two COMT genes, and two CCR genes) and MYB, NAC, ERF, and TCP transcription factor genes were involved in lignin metabolism. An analysis of RT-qPCR confirmed that these DEGs were involved in the biosynthesis and regulation of lignin. These findings further help us understand the mechanisms of lignin biosynthesis and provide a theoretical basis for peel color control and quality improvement in pear breeding and cultivation.

## 1. Introduction

Peel color is a significant trait of agronomic characteristics of fruit and a key quality factor in determining the price of commodities. Based on different peel colors, pears can be divided into three groups of green pears, brown pears, and red pears, respectively [1]. Green pears and brown pears are the two main pear groups cultivated in China. Consumers prefer brown-peeled pears with consistent color [2]. Studies have shown that in the early stage of fruit development, the stomatal guard cells under the epidermis break down to form holes, and the parenchyma cells in the holes became lignified and protrude from the epidermis and cuticle to form fruit points. The fruit points continue to be shaped by lignification, and brown fruit skins are formed with the accumulation of cork formation. Lignin is an important component of the cork layer, so the accumulation of lignin may be related to the formation of brown pear peel [3,4,5,6]. However, there are few reports regarding the molecular mechanisms of lignin biosynthesis in sand pear peel.

Lignin is one of the main components of plant cell walls and is a phenolic polymer. The biosynthesis of lignin includes the phenylalanine metabolic pathways and lignin-specific pathways [7]. Many enzymes play a key role in lignin biosynthesis, for instance, phenylalanine ammonia lyase (PAL), cinnamate 4-hydroxylase (C4H), 4-coumarate: CoA ligase (4CL), cinnamoyl-Co A reductase (CCR), caffeoyl-CoA O-methyltransferase (CCoAOMT), ferulate-5-hydroxylase (F5H), caffeic acid O-methyltransferase (COMT), and cinnamyl alcohol dehydrogenase (CAD) as well as peroxidase (POD) and laccase (LAC). These enzymes participate in the biosynthesis of monolipids and the polymerization of monomers to produce lignin polymers [8,9,10,11,12,13,14].

The available literature indicates that transcription regulators, such as myeloblastosis transcription regulators (MYBs), basic helix-loop-helixes (bHLHs), ethylene responsive factors (ERFs), NACs (NAM, ATAF1/2, and CUC1/2–containing transcription factor) are involved in the regulation of lignin biosynthesis [15,16,17,18]. MYB family members regulate lignin biosynthesis by interacting with AC elements [AC-I (ACCTACC), AC-II (ACCAACC), and AC-III (ACCTAAC)] in the promoter regions of key genes during phenylpropane biosynthesis [19,20]. Studies show that *PtrMYB120* is involved in the regulation of lignin biosynthesis in poplar [19,21]. *PbMYB140* is responsible for activating the expression of genes involved in the biosynthesis of lignin, while its own expression is regulated by *PbMYB83* [22]. *PbrMYB24* regulates the biosynthesis of lignin and cellulose in stone cells by binding different *cis*-acting elements in pears [23]. The overexpression of *SHINE* (*AP2/ERF* gene) in the rice of *Arabidopsis* could reduce the content of lignin by 45% [24]. Another study showed that the transcription factors *ERF18/34/35* from AP2/ERF family genes could regulate stem growth and wood properties in poplar [25]. *DcERF1* and *DcERF2* could bind to the GCC box of the promoter of *DcPAL3* [26]. Furthermore, recent research found that the overexpression of *PpERF1b* in fruit and stably in callus increased lignin accumulation and the expression levels of lignin biosynthesis genes in pear callus, leading to the formation of ‘hard-end’ pear fruit [27]. The above results indicate that *AP2/ERF* genes have a potential function in the regulation of lignin biosynthesis.

Previous studies regarding lignin biosynthesis in pears mainly focus on the formation of stone cells in the pulp [28,29,30]. Some studies related to lignin metabolism have investigated pigmentation and russeting formation in pears; however, it is yet unknown how lignin contributes to the formation of peel color. The studies regarding lignin-related color formation in peel mainly focus on the resolution of budburst varieties with a brown peel, such as the ‘Dangshansu’ pear and the budburst variety ‘Xiusu’ pear, both of which have a green peel within 100DAF. It is also found that 75DAF to 125DAF is the key period for brown peel formation in the ‘Xiusu’ pear [31]. Lignin content is consistently higher in the ‘Xiusu’ pear than in the ‘Dangshansu’ pear at 75–125DAF. Increased lignin content is positively correlated with the expression levels of *PAL2*, *4CL1*, *CAD1*, and *POD4* in the ‘Xiusu’ pear, whereas its content is only positively correlated with the expression level of *POD4* in the ‘Dangshansu’ pear [32]. Using ‘Huanghua’ (brown peel pear) and its bud mutation green peel variety ‘Huanghua’ (green peel pear) as materials, Lv et al. found that the expression level of *HHT* was higher in brown peel pears (‘Huanghua’) than in green peel ones (‘Huanghua’) at different periods of fruit development [33]. A transcriptomic analysis showed that the expression levels of *POD* and *4CL* in the phenylpropanoid synthesis pathway, as well as *HHT1* and *CYP86A1* in the keratin/thrombin and wax synthesis pathways, were suppressed in the peel of ‘Sucuiyihao’ (green peel with few rust dots), which might lead to a reduction in fruit rust in ‘Sucuiyihao’ [34].

Lignin content substantially affects the appearance and texture of pear peel, determining the commodity attribute of pear fruits. In order to unveil the mechanisms underlying brown peel formation in pears, we used the newly bred brown peel pear variety ‘Xinyu’ as the material and investigated the dynamic of lignin and the regulation details of its metabolism during the development of the pear based on biochemistry, metabolomic and transcriptomic methods. RNA-seq, RT-qPCR, and biogenic analyses were used to characterize the expression patterns of lignin-metabolism-related genes in the peel of the ‘Xinyu’ pear. Furthermore, the key genes and main regulatory factors of lignin biosynthesis were screened and identified by using correlation analysis and co-expression analysis, respectively. Our results could provide the theoretical basis for genetic breeding and quality improvement of brown-peel pears.

## 2. Results

### 2.1. Changes in Lignin Contents during Fruit Development of ‘Xinyu’ Pear

The appearance of the ‘Xinyu’ pear at the different development stages is presented in Figure 1. The color of the peel develops from an emerald green to a brown-green color, and finally to brown from 20DAF to 100DAF. The contents of lignin in the ‘Xinyu’ pear were measured by a UV-visible spectrophotometer (Figure 2). The lignin content was low at 20DAF and slowly increased to a maximum value at 50DAF in the peel of the ‘Xinyu’ pear, then gradually decreased from 60DAF to 100DAF (two-tailed *t*-test, *p* = 9.59 × 10^−3^). At 100DAF, the content of lignin eventually dropped to the same level as the beginning of fruit development (20DAF) (Figure 2). Overall, the ‘Xinyu’ pear had relatively high contents of lignin in the early development stages and declined as the fruit developed.

### 2.2. Observation of Fruit Peel by Scanning Electron Microscope (SEM) in ‘Xinyu’ Pear

The process of fruit spot formation and the structure of the cork layer in the peel were characterized by an SEM. As shown in the pictures, complete pores could be seen on the surface of the pear peel, and the wax film on the peel was relatively smooth with small gaps at 20DAF (Figure 3). The peel began to form fruit spots, where the cork in the cells burst through the surrounding epidermis and gradually filled the cavity, but the fruit surface was still smooth. The fruit spots gradually increased and unevenly formed a rough cuticle film and suberization. It was difficult to observe the pores at 50DAF. There was extensive damage to the fruit surface, where intercellular connections by cell walls were completely disrupted and gaps of varying sizes formed at the later stages of fruit development (Figure 3).

### 2.3. Metabolomic Analysis of the ‘Xinyu’ Pear during the Development of Fruit

To further confirm the metabolite changes in the peel of the ‘Xinyu’ pear during the peel color change process, a lignin-related metabolomic assay was carried out using UPLC-ESI-MS/MS at nine developmental stages. We detected nine lignin-related metabolites in 27 samples, including L-phenylalanine, p-coumaric acid, counmaic aldehyde, caffeic aldehyde, caffeyl alcohol, ferulic acid, sinapyl alcohol, sinapaldehyde, and conifer aldehyde. The first three highest contents of these identified lignin-related metabolites were L-phenylalanine, p-coumaric acid, and conifer aldehyde, respectively (Table 1). A principal component analysis (PCA) and cluster analysis showed that these metabolites were classified into three groups, named Group I, Group II, and Group III. Group I contained p-coumaric aldehyde, ferulic acid, sinapaldehyde, and sinapyl alcohol, showing a relatively high level of accumulation from 30DAF to 70DAF. Group II contained only L-phenylalanine, which had the highest level at 20DAF, rapidly decreased at 30DAF and 40DAF, and then recovered at 70–90DAF. Group III contained p-coumaric acid, conifer aldehyde, caffeic aldehyde, and caffeyl alcohol, which showed a higher level from 20DAF to 70DAF, and gradually decreased from 80DAF. Except for L-phenylalanine, the other metabolites obviously showed a developmental specificity and had a higher content at the early development stage of pears. However, L-phenylalanine showed the highest level in the peel among these nine metabolites during the development of the ‘Xinyu’ pear (Figure 4).

### 2.4. Transcriptome Analysis of the ‘Xinyu’ Pear during the Development of Fruit

A total of 27 libraries were established using peel samples from nine development stages with three biological replicates for the ‘Xinyu’ pear. After removing the adaptor and low-quality sequences, an average of 6.43 GB of sequencing data was obtained from 27 samples. After filtering the raw data, 1,159,872,674 clean reads were obtained, the length of which ranged from 38,688,270 to 47,956,942 (Appendix A). A correlation analysis of expression levels (represented by Transcripts Per Million, TPM) of identified genes showed a high degree of correlation among different biological replicates of the same developmental stage (Appendix A), suggesting that there was a good reproducibility among different biological replicates.

The differentially expressed genes (DEGs) were identified according to the criteria of |log2(Fold Change)| ≥ 1 and a threshold of false discovery rate of ≤0.01. Finally, a total of 28,293 genes were obtained among those 27 samples (Appendix A). Two distinct phases in the peel of the ‘Xinyu’ pear could be distinctly divided during fruit development, namely, the earlier fruit development stages (20DAF, 30DAF, 40DAF, 50DAF, and 60DAF) and the post-development stages (70DAF, 80DAF, 90DAF, and 100DAF). According to the expression patterns of these DEGs during the development of the ‘Xinyu’ pear, these DEGs were clustered into two distinct groups. DEGs of group I (56.17%, 15,891 out of 28,293) were specifically expressed during the earlier stages of fruit development (Figure 5A), whereas DEGs of group II (43.83%, 12,402 out of 28,293) had higher expression levels at the later stages of fruit development (Figure 5A). The PCA and cluster dendrogram of the transcriptomic data further confirmed the classification of DEGs into these two groups (Figure 5B,C).

### 2.5. Analysis of Unigenes Related to Lignin Biosynthesis Pathways in Fruit Peel

To further understand the mechanism of peel color formation in the ‘Xinyu’ pear, the expression patterns of structural genes related to lignin metabolism were analyzed. Based on KEGG annotation, a total of 148 DEGs were found to be involved in the metabolism pathway of lignin (Appendix A). Structural genes annotated as *PAL*, *C4H*, *4CL*, *HCT*, *CSE*, *CCoAOMT*, *COMT*, *CCR*, *CAD*, *POD*, and *LAC* had different expression patterns during the fruit development of the ‘Xinyu’ pear (Figure 6). The expression levels of most genes were relatively low in fruit peel at nine developmental stages of the ‘Xinyu’ pear. Pathway analysis showed that there were ten structural genes (named *PAL*, *C4H*, two *4CLs*, *HCT*, *CSE*, two *COMTs*, and two *CCRs*) involved in the accumulation of lignin, which might play important roles in the formation of fruit spots and the peel color of the ‘Xinyu’ pear (Figure 1 and Figure 3).

### 2.6. Co-Expression Analysis of Genes Related to Lignin Synthesis Pathway

In order to explore the correlation between genes and distinguish the co-expression genes, 12,304 DEGs (coefficient of variation > 50%) were selected to perform a weighted gene co-expression network analysis (WGCNA) in the peel transcriptome of the ‘Xinyu’ pear at nine developmental periods (Appendix A). A total of nine gene co-expression modules were identified (Figure 7A). A heatmap of module-trait correlations was constructed by combining the accumulation levels of phenylalanine pathway metabolites at nine developmental stages in the ‘Xinyu’ pear (Figure 7B). Among them, the green and green-yellow modules contained significantly more genes than other modules, accounting for 31.26% and 37.78% of the total DEGs, respectively. The sky blue, black-gray, and pink modules mainly contained DEGs that were highly expressed during the later stages of fruit development. Results showed that the green-yellow module was positively correlated with p-coumaric acid, ferulic acid, sinapyl alcohol, and conifer aldehyde in the phenylalanine pathway of ‘Xinyu’ pear peel, with the accumulation level of conifer aldehyde being extremely positively correlated with the green-yellow module. In the correlation network of the green-yellow module, 10 structural genes via prediction from the correlation of the accumulation levels of metabolites related to the phenylalanine metabolic pathway, including *PAL1* (EVM0009710), *4CL2* (EVM0025898), *4CL11* (EVM0004767), *C4H2* (EVM0004300), *HCT3* (EVM0017088), *CSE1* (EVM0005577), *COMT1* (EVM0017121), *COMT2* (EVM0039159), *CCR1* (EVM0013784), and *CCR12* (EVM0021785), were considered as the hub genes. The expression levels of these structural genes were strongly correlated with the contents of p-coumaric acid, ferulic acid, sinapyl alcohol, and conifer aldehyde in the ‘Xinyu’ pear. The transcripts of 76 hub genes of transcription factors were identified in the green-yellow module (Appendix A). Among them, the expression levels of ten *MYBs*, four *NACs*, thirteen *ERFs*, and six *TCP* transcription factor genes were closely related to the phenylalanine metabolic pathway. Gene network analysis showed that eight *MYB* transcription factor genes simultaneously regulated the expression levels of *PAL1*, three *NAC* transcription factors simultaneously regulated the expression levels of *HCT3*, eight *ERF* transcription factors simultaneously regulated the expression levels of *4CL11*, and four *TCP* transcription factor genes simultaneously regulated the expression levels of *COMT2*, respectively (Figure 7C).

### 2.7. RT-qPCR Validation of the Results of Transcriptomic Data in ‘Xinyu’ Pear Peel

In order to validate the reliability of transcriptomic data and the results of the gene co-expression network analysis, closely related structural genes and transcription factors involved in the phenylalanine metabolic pathway were submitted for RT-qPCR analysis. Our results showed that the expression patterns of eight structural genes and seven transcription factor genes related to the L-phenylalanine metabolic pathway were highly consistent with the dynamics of the corresponding gene obtained by transcriptome analysis (Figure 8). The structural genes, such as *C4H2*, *CCR1*, *CCR3*, *COMT1*, *COMT2*, *HCT3*, *PAL1*, and *PAL4*, showed the highest expression levels at 30DAF–40DAF, then their expression levels gradually decreased along with fruit development. The transcription factors, such as *ERF74*, *ERF93*, *ERF66*, and *ERF91*, had relatively higher expression levels from 20DAF to 40DAF, then rapidly decreased after 40DAF (Figure 8J–M). *ERF15* reached its highest expression at 30DAF and then presented a declining expression pattern. *MYB11* increased rapidly from 20DAF to 50DAF and reached its highest level at 50DAF–70DAF. *MYB179* was only expressed before 70DAF in the fruit development of the ‘Xinyu’ pear, and had an up-down expression pattern, containing two peaks at 40DAF and 60DAF, respectively. The high consistency between the RT-qPCR analysis and transcriptomic data of all these candidate genes further verified that the RNA-Seq results are reliable to identify the DEGs in the ‘Xinyu’ pear during different development stages.

## 3. Discussion

Peel color is an important appearance trait of pears, which affects its commodity value. In production, the peel of sand pear is mainly green or brown; however, due to internal and exterior environment factors, uneven fruit rust often develops on the surface of green pears, resulting in a decline in the appearance of the fruit. As a market selection factor, pears with uniformly colored peel, such as brown pears, were more favored by consumers [2]. Therefore, the breeding of brown pears is the main objective of pear breeding. Previous studies have shown that brown peel is related to the formation of the cork layer [35]. The broken cuticle membrane, together with fruiting spot detachment and lignification of the epidermal cells leads to the formation of the cork layer, resulting in the production of irregular brown fruit rust on the surface (Figure 3). In these processes, the biosynthesis and accumulation of lignin is one of the key factors for the formation of the cork layer [35]. In this study, SEM as well as the measurement of lignin content revealed that the stomatal margins cracked at 30DAF, and then the content of lignin significantly increased and the fruit spots began to appear (Figure 2 and Figure 3). The epidermis was broken and underwent suberization at 40DAF. In the meantime, the peel showed further expanding rust spots and the highest content of lignin. The peak of lignin content (50DAF) appeared before the peak of the fruit rust area (60DAF), indicating that the formation of brown fruit rust was closely related to the biosynthesis, transportation, and deposition of lignin (Figure 3). Similarly, Shi et al. reported that green light culture decreased the enzymatic activities and the expression levels of enzymes involved in lignin biosynthesis, leading to less accumulation of lignin and less brown fruit rust in green peel cultivar ‘Cuiguan’ pear peel [36].

The available literature has shown that an increased content of lignin involves the phenylalanine metabolic pathway [37]. There are many types of lignin in plants, including H-lignin, G-lignin, and S-lignin. Among them, G-lignin and S-lignin are the dominant types of lignin in pear fruits [38,39]. P-coumaric acid is a key substrate for lignin biosynthesis. Ferulic acid and pine aldehyde are the key rate-limiting metabolites for the biosynthesis of G-lignin and S-lignin, and mustelol is a monomer substrate for the synthesis of S-lignin [37]. In this study, metabolomic measurement revealed the presence of several compounds related to lignin biosynthesis in ‘Xinyu’ pear fruit peel. These included L-phenylalanine, phenolic acids (p-coumaric acids and ferulic acids), phenolic aldehydes (coumaric aldehydes, conifer aldehyde, caffeic aldehydes, and sinapaldehyde), and phenolic alcohols (caffeyl alcohol and sinapyl alcohol) (Table 1). Among them, L-phenylalanine, p-coumaric acid, and pinacolone aldehydes are the top three substances with the highest content, which suggested that the lignin metabolism pathway of the brown peel of pears was mainly dominated by the production of G-lignin and S-lignin [37,39]. This is because L-phenylalanine is a primary metabolite and is involved in the biosynthesis of a variety of secondary metabolites. The content of L-phenylalanine is affected by a variety of factors. Here, we showed that the dynamic of L-phenylalanine in ‘Xinyu’ pear peel was obviously different from other metabolites (Table 1). The conversion of p-coumaric acid to caffeic acid in the phenylalanine metabolic pathway is an essential process for the synthesis of G-lignin and S-lignin in plants, which occurs mainly through two pathways: the first one is that p-coumaric acid is converted directly into caffeic acid by the catalytic action of the C3′H enzyme, and the second one is that p-coumaric acid is ultimately converted to caffeic acid via p-coumaric coenzyme A, p-coumaroylmanganinic acid, and caffeoylmanganinic acid, subsequently [39]. The regulatory network of phenylalanine metabolism constructed in this study contained one *HCT* gene and one *CSE* gene, and did not contain the *C3′H* gene, suggesting that p-coumaric acid might be converted to caffeic acid and thus affected the biosynthesis of G-lignin and S-lignin through the pathways via p-coumarin coenzyme A, p-coumaroylmangiferic acid, and caffeoylmangiferic acid during lignin biosynthesis during the fruit development of the ‘Xinyu’ pear (Figure 6).

The *PAL* gene encodes a phenylalanine ammonia cleavage enzyme that catalyzes the conversion of L-phenylalanine to cinnamic acid, the initial step of the lignin metabolic pathway [39]. Here, the highest expression levels of *PAL1* and *PAL4* were observed at the time when the fruit spots started to turn brown (40DAF), suggesting that the expression level of *PALs* reached a certain level in order to catalyze the precursor substances, thus promoting the synthesis and accumulation of lignin (Figure 8). This is consistent with the previous results obtained in other pear cultivars [40]. WGCNA showed that the expression level of *PAL* was highly positively correlated with the dynamics of ferulic acid and turpentine aldehyde during the development of the ‘Xinyu’ pear (Figure 7C). In conclusion, the above results suggest that *PAL1* and *PAL4* might play vital roles in the biosynthesis and metabolism of lignin during the formation of brown peel in the ‘Xinyu’ pear. Generally, the high expression level of *C4H* genes was associated with a high lignin content in plants [41]. Previous studies showed that *PbC4H1* was highly expressed only during the ripening stage of pear fruit, *PbC4H2* was involved in cell wall development and lignin biosynthesis, and the expression pattern of *PbC4H3* was correlated with stone cell content, suggesting that *PbC4Hs* might play important roles in lignin biosynthesis and stone cell development in pear [28,29,42]. In the current study, the highest expression level of *C4H2* was only observed in the early development stage of the ‘Xinyu’ pear (30DAF), and then gradually decreased along with the development of the pear (Figure 8A). A previous study indicated that there were 31 *CCR* genes in white pears, of which *PbCCR1*, *PbCCR2*, and *PbCCR3* were up-regulated in the early development stage of pear fruit (55DAF–70DAF) and were positively correlated with the content of lignin and the number of stone cells, while the other CCR genes were weakly expressed [43]. In this study, *CCR1* and *CCR3* were up-regulated during the early developmental period (20DAF–40DAF) in the peel of the ‘Xinyu’ pear, similar to the changing pattern of lignin content, suggesting that they might play an important role in pear lignin biosynthesis. It has been shown that the lignin content in the *hct* mutant of *Arabidopsis* was significantly lower than that in a control plant [44]. Another study demonstrated that poplar trees homozygous for a recessive allele expressing a truncated HCT1 protein result in a modified lignin composition and a significant increase in H-lignin content [45]. COMT was one of the key enzymes in lignin biosynthesis and might potentially regulate the synthesis of S-lignin in the lignin biosynthesis pathway [46]. Down-regulation of *COMT* genes can reduce the biosynthesis and content of lignin in *Arabidopsis* [47]. In this study, one *HCT3* gene and two *COMT* genes were isolated and their expression levels during different development stages of the ‘Xinyu’ pear were studied. Results show that both of them had the highest expression levels at 40DAF, then decreased gradually at the later development stages of the ‘Xinyu’ pear. Such trends were consistent with the dynamic of lignin contents during fruit development (Figure 2 and Figure 8).

To explore the regulatory mechanism of lignin biosynthesis in fruit peel, 76 regulatory genes including MYB, NAC, ERF, and TCP transcription factors were screened and identified in the metabolic regulatory network (Appendix A). Genome-wide identification and evolution research showed that 28 out of 104 isolated MYB transcription factor genes encoded R2R3 subfamilies, which played a regulatory role in the lignin biosynthesis in the Chinese white pear [48]. The *PbMYB24*, *PbMYB61*, *PbMYB80*, *PbrMYB169*, and *PbMYB308* genes could activate lignin biosynthesis in fruit and potentially regulate stone cell lignification in white pears [17,23,49]. *PbMYB140* had a significant difference in gene expression between the brown-skinned pear ‘Xinggao’ and its green budburst variety ‘Daguoshuijing No.l’, indicating that it was possibly involved in the regulation of lignin synthesis in the formation of pear fruit rust [22]. In this study, we screened and identified 10 MYB transcription factors which had the potential to bind to the AC *cis*-elements of structural genes involved in the phenylalanine metabolic pathway and co-expressed with these structural genes. Among 10 MYB transcription factors, eight MYB genes regulated the expression of the PAL1 gene together, suggesting their role in the regulation of the phenylalanine metabolic pathway and thus lignin biosynthesis in the fruit peel of the ‘Xinyu’ pear (Appendix A; Figure 7C). ABA signaling activates the phosphorylation of AtNST1, a member of the NAC transcription factor family. This phosphorylation allows AtNST1 to acquire the function of regulating downstream genes and increase cellulose and lignin accumulation in *Arabidopsis* [50]. Over-expression of *EgNAC141* increased the biosynthesis of lignin and led to a higher content of lignin in stems and xylem in transgenic *Eucalyptus* seedlings than the control ones [51]. *PpNAC187* was reported to be involved in the formation of hardhead fruit by inducing lignification in pears [52]. In the current study, we screened and identified four NAC transcription factor genes. Three of these NAC transcription factor genes co-regulated the *HCT3* gene (Appendix A; Figure 7C). The *EjERF39* gene cooperated with the *EjMYB88* gene to regulate pulp lignin content and promote fruit lignification under low-temperature conditions in loquat [53]. Over-expression of the desert moss ERF transcription factor family genes promoted lignin accumulation and enhanced *Verticillium dahliae* resistance in *Arabidopsis* [54]. In contrast, the *PagERF81* gene was found as a negative regulator of lignin biosynthesis, whose knockdown led to an increased lignin content and whose over-expression led to a decreased lignin content in plants [55]. Transient over-expression of the *PpERF1b*-*like* gene increased the content of lignin and the expression levels of some other lignin biosynthesis genes in pear fruit [27]. In the current study, a total of five *ERF* genes were expressed at a higher level in the peel during the early stage of fruit development compared with those expressed at the later development stage of the ‘Xinyu’ pear (Appendix A). WGCNA results showed that these five *ERF* genes were involved in the regulation of the structural gene *4CL11*, which might participate in the regulation of lignin content in pear fruit peel (Figure 7C).

## 4. Material and Methods

### 4.1. Plant Materials

The ‘Xinyu’ pear was certified with newly bred plant variety rights in 2018 and gained the approval of the Fujian Provincial Forest Variety in 2022 (Certificate No. S-SV-PP-017-2022). The fruit peel of a mature “Xinyu” pear is bright yellow-brown, and it was categorized as the brown-peel pear. The ‘Xinyu’ pear trees used in this study were grown in Jianning county, Sanming city, Fujian Province, China, receiving regular agricultural measures and the prevention of pests and diseases. Fruit samples were picked at 20, 30, 40, 50, 60, 70, 80, 90, and 100 days after flowering (DAF) from three trees on a clear day, respectively (Figure 1). Three biological replicates were taken for each fruit sample. All the trees were grown in a square of 4 m × 5 m and received standard horticultural practices such as irrigation, sod cultivation, and pest/disease prevention in a rainproof greenhouse facility. Samples were collected during a sunny noon, peeled with a paring knife (thickness < 1 mm), and sliced into appropriate pieces. In all cases, there were three biological replicates for each sampling date and each biological replicate was mixed from at least five fruits. The peel pieces from different pear fruits were wrapped into one aluminum foil bag and labeled. Then, the samples were frozen in liquid nitrogen and stored at −80 °C until assay.

### 4.2. Lignin Extractions and Analysis by Spectrophotometer

Lignin was extracted from the peels and analyzed by a spectrophotometer according to the instructions of the lignin assay kit (G0708W, Suzhou Grace Bio-technology, Suzhou, China). Briefly, peel samples were oven-dried to constant weight at 80 °C; then the dry samples were ground into a fine powder and sieved using a 0.425 mm mesh sieve. About 1.5 mg of powder sample was extracted using 1.5 mL of 80% ethanol and was mixed well by vortex oscillation mixing; then, it was put into a 50 °C water bath for 20 min. After water cooling, the mixture was centrifuged at 12,000× *g* for 10 min at 25 °C. The precipitation was collected and extracted using one milliliter of 80% ethanol in a 50 °C water bath for 20 min. The mixture was again centrifuged at 12,000× *g* for 10 min at 25 °C. The precipitate was dried at 95 °C and then used for lignin measurement using the lignin assay kit. An aliquot of 200 μL of the reaction mixture was placed in a 96-well UV plate to determine the absorbance at 280 nm. There were three replicates for each measurement of lignin.

### 4.3. ‘Xinyu’ Pear Peel at Different Periods of Fruit Growth and Development Were Observed by Scanning Electron Microscope (SEM)

The structure of the peel surface of the ‘Xinyu’ pear at the different development stages (20DAF–100DAF) was examined with a scanning electron microscope (100×, 150×, 500×) (JEOL JSM-6380, JEOL Ltd., Tokyo, Japan). When the peel samples were collected in the field, the peel samples were cut into 0.5 cm × 0.5 cm slices with a scalpel blade and were immediately put into a 25% glutaraldehyde fixative, taken back to the laboratory, and stored in the refrigerator at 4 °C. The slices were rinsed several times with a phosphate buffer before preparation, dehydrated by ethanol gradient, then supercritically dried, and sprayed with gold powder. Finally, the shape and surface morphology of the fruit dots on ‘Xinyu’ pear peels were observed and photographed via SEM.

### 4.4. Metabolomic Profiling Analysis

The lignin metabolome was performed by Metware Biotechnology Co., Ltd. (Wuhan, China) using the UPLC-MS/MS method following their standard metabolic operating procedures. Briefly, 100 mg of freeze-dried peel powder was extracted with 1 mL of 70% aqueous methanol overnight at 4 °C. The mixture was centrifuged for 10 min at 10,000× *g*, and then the supernatants were filtered with 0.22 μm pore size membrane filters. Based on the self-built database and the public database of metabolite information, the test samples were examined using the multi-reaction monitoring model (MRM). The chromatographic peaks were analyzed using the Analyst software (Version 1.6.3) and obtained qualitative and quantitative results of metabolites. The lignin metabolites were annotated using KEGG, MWDB, MassBank, KNAPSAcK, and HMDB, and metabolites with |Fold Change| ≥ 2 or ≤0.5 were selected and identified as the final differential metabolites. A clustering heat map and differential metabolite correlation analysis were visualized using the pheatmap package in the R language environment.

### 4.5. Transcriptome Analysis

The total RNA was extracted from peel samples of the ‘Xinyu’ pear using the TRIzol reagent (Invitrogen, Carlsbad, CA, USA), and DNase I was then used to remove genomic DNA (Takara, Dalian, China). The quality, quantity, and integrity of the total RNA were evaluated using the Aglient 2100/LabChip GX (Agilent Technologies, Santa Clara, CA, USA). The sequencing library construction of samples was generated using a TruSeq RNA Sample Preparation Kit (Illumina, San Diego, CA, USA), and the library preparations were sequenced on the Illumina NovaSeq6000 platform. The reference genome database and gene annotation files were extracted from the ‘Cuiguan’ pear genome database (https://www.rosaceae.org/Analysis/11815273?pane=bio_data_2_rsc_assembly, accessed on 23 January 2024). The filtered reads were aligned to the reference genome using StringTie (Version 2.2.1) [56]. Transcripts Per Million (TPM) was used as the quantification unit of gene expression levels [57]; cluster dendrogram and heatmaps were drawn in the R language environment (Version 4.2.1) using the base package and the pheatmap package. The transcriptomic data are presented in Appendix A.

### 4.6. Co-Expression Network Construction of Metabolome and Transcriptome

Differentially expressed genes [coefficient of variation (CV) > 0.5] were selected as the base set for the weighted gene co-expression network analysis by excluding genes that were not detected or had low relative expression (average TPM < 1), and the profile of co-expression networks was generated using the WGCNA package (Version 1.72) in the R language (Version 4.2.1) environment [58]. The parameters were set as follows: the construction of co-expression modules was performed using the network constructor (block-wise modules) with default parameters, the soft threshold used was set to seven, the TOM matrix type (TOM type) was signed, the merge cut height was set to 0.25, and the minimum module size (min module size) was set to 100. Gene trait values for each module were used to identify the relationship of the module to lignin metabolite content at the different development stages of the ‘Xinyu’ pear. The transcriptional *cis*-element binding site prediction in the promoter region of lignin metabolism-related structural genes (1500 bp upstream to 200 bp downstream region of the transcription start site) was predicted using the online software FIMO (https://meme-suite.org/meme/doc/fimo.html, accessed on 7 March 2024) [59]. The position frequency matrices (PFMs) of transcription factors used in the prediction were downloaded from the plantTFDB database [60]. The transcription regulatory networks were generated by combining Pearson correlation coefficients between structural genes and transcription factors and the prediction of *cis*-element binding sites in the promoter regions of structural genes in the same module. Key modules associated with the lignin metabolism in ‘Xinyu’ pear peel and the co-expression network were visualized for subsequent analysis using the Cytoscape_V.3.7.1 software (MAC, Fresno, CA, USA) (Appendix A) [61].

### 4.7. RT-qPCR Analysis

To validate the reliability of the transcriptome data, eight putative candidate functional genes and seven transcription factors were selected from the DEGs data and further analyzed using RT-qPCR analysis. The total RNA of nine peel samples was extracted using the TRIzol^®^ Reagent (Invitrogen, USA). RT-qPCR analysis was performed on a CFX96 real-time system (Bio-Rad, Hercules, CA, USA) with a TaKaRa 2× SYBR Premix ExTaq^TM^ kit (Takara, Dalian, China). RT-qPCR analysis was performed using the following steps: 95 °C for 30 s, followed by 40 cycles at 95 °C for 5 s and 30 s at 72 °C. The relative expression level was analyzed using the 2^−ΔΔCT^ method.

The expression level of each gene was normalized by the same gene of the 20DAF sample, whose expression level was set to 1. The acting gene was used as the internal reference gene. The RT-qPCR primer sequences are listed in Appendix A. There were three biological and three technical replicates used for the RT-qPCR assays.

## 5. Conclusions

The newly bred pear cultivar ‘Xinyu’ is favored by customers in southern China due to its uniform brown peel. Previous research has shown that the biosynthesis and accumulation of lignin are involved in the formation of fruit spots and cork layers, which leads to the appearance of brown colored peel in sand pears. In this study, we investigated the dynamic of fruit surface, lignin-related metabolites, and expression levels of genes related to lignin metabolism during nine development stages of the ‘Xinyu’ pear using SEM, metabolomics, and transcriptomics. Our results showed that along with the highest contents of lignin appearing during 30DAF to 70DAF, the fruit spots gradually increased and unevenly formed a rough cuticle film and suberization. WGCNA, correlation analysis, and gene network analysis demonstrated that the contents of key metabolites in lignin biosynthesis such as p-coumaric acid, ferulic acid, and sinapyl alcohol, were regulated by the expression levels of lignin metabolism-related structural genes, such as *PAL*, *4CL*, *C4H*, *HCT*, *CSE*, *COMT*, and *CCR*. Meanwhile, transcription factor family genes *MYB*, *NAC*, *ERF*, and *TCP* also participated in the lignin biosynthesis process. These findings highlight the key genes and metabolites involved in the lignin biosynthesis process in brown-peel sand pears, which are of great importance in unveiling the physiological and genetic mechanisms determining the product appearance of fruit, and provide a theoretical basis for pear color control and quality improvement in sand pears.

## Figures and Tables

**Figure 1 ijms-25-07481-f001:**
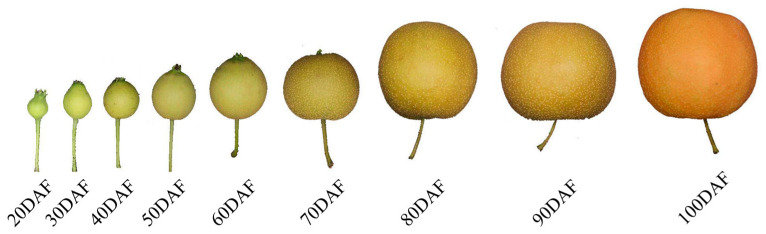
The appearance of the ‘Xinyu’ pear at different stages of fruit development.

**Figure 2 ijms-25-07481-f002:**
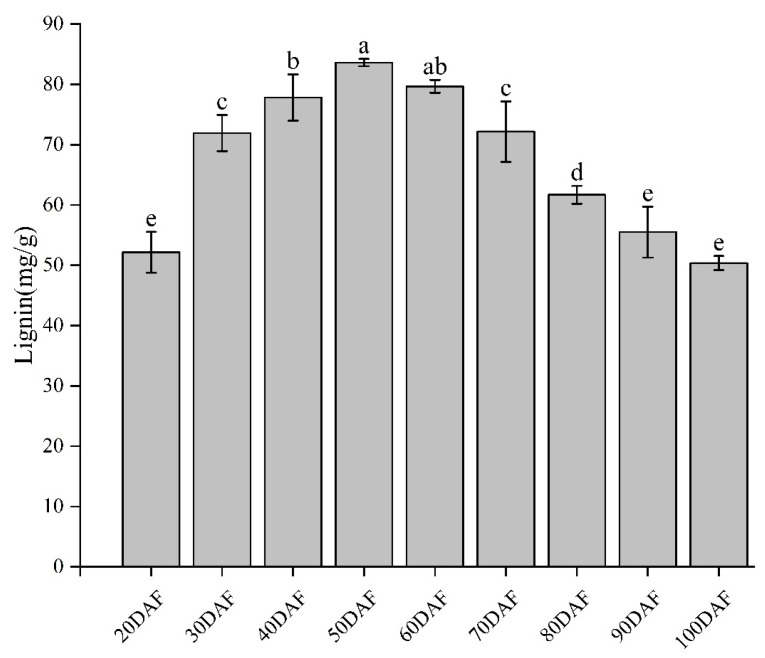
Lignin contents in the peel of the ‘Xinyu’ pear at the different development stages of fruit. Bars represent means ± standard deviation (n = 3). Differences among the samples at nine development stages were analyzed by multiple comparisons of a one-way ANOVA (Fisher method) with three biological replicates. The bars sharing the same letter do not have a significant difference from each other, but any two bars with different letters have a significant difference with *p* < 0.05.

**Figure 3 ijms-25-07481-f003:**
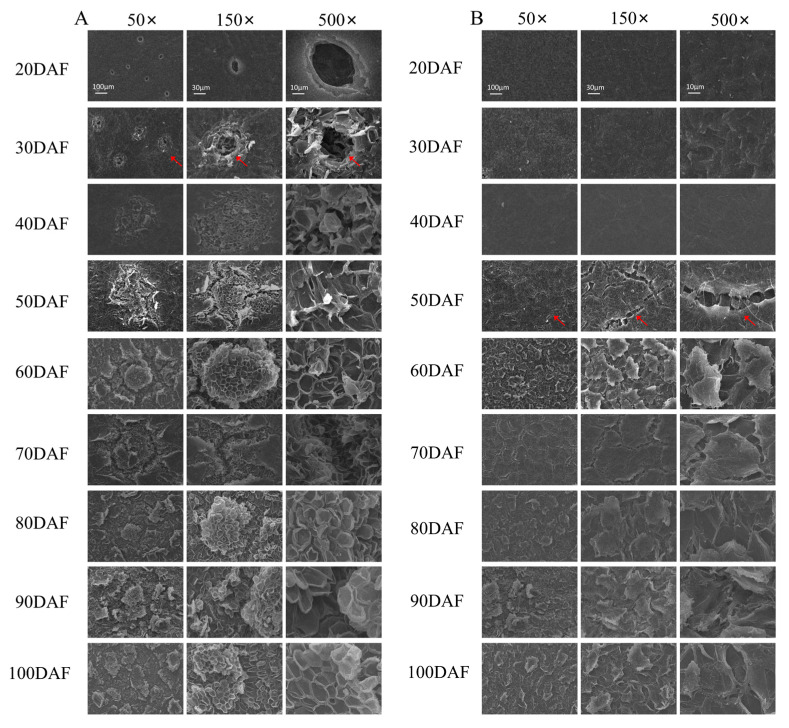
Scanning electron microscopy (SEM) of the peel of the ‘Xinyu’ pear at different stages of fruit development. (**A**): Fruit pores; (**B**): fruit surface. (**A**): The red arrows indicate the period when the fruit spots began to form. (**B**): The red arrows indicate the period when the fruit surface began to crack. The bars in 20DAF are shared by each single drawing per column.

**Figure 4 ijms-25-07481-f004:**
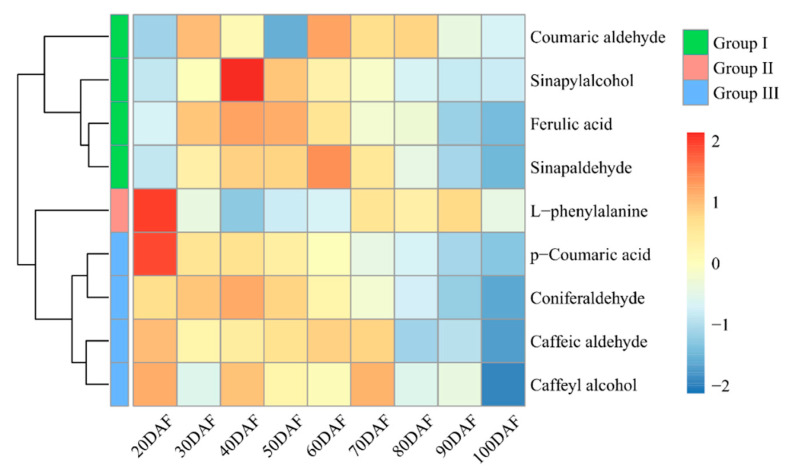
A heatmap of nine different lignin-related metabolites in the peel at different developmental stages of the ‘Xinyu’ pear. The contents of lignin-related metabolites were visualized following the respective normalized value (n = 3). The color scale indicates the value of normalization content for each metabolite, with blue representing a lower content and red representing a higher content.

**Figure 5 ijms-25-07481-f005:**
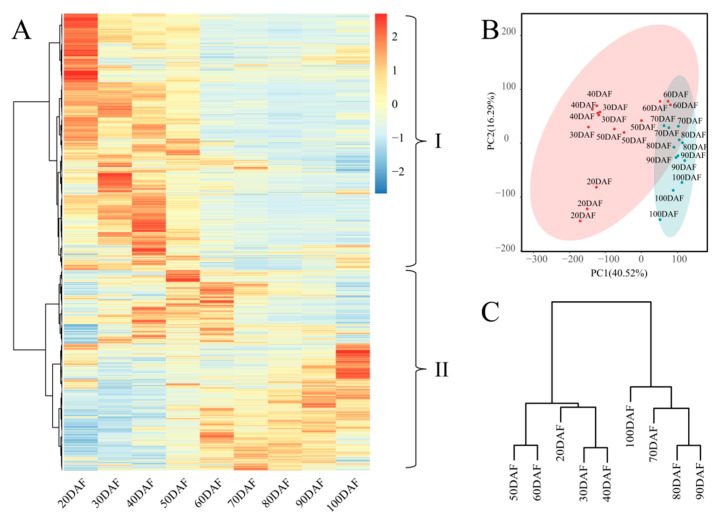
The transcriptome sequencing data at the different stages of fruit development of the ‘Xinyu’ pear. (**A**): A heatmap of 28,293 expression genes that were divided into two groups at 9 developmental stages of ‘Xinyu’ pear fruit peel. Group I was highly expressed during the early fruit development stages (20DAF–60DAF) and group II was highly expressed during the later fruit development stages (70DAF–100DAF). Different colors indicate the expression levels of DEGs, from blue (low) to red (high); (**B**): PCA of 28,293 expression genes; and (**C**): cluster dendrogram of 28,293 expression genes.

**Figure 6 ijms-25-07481-f006:**
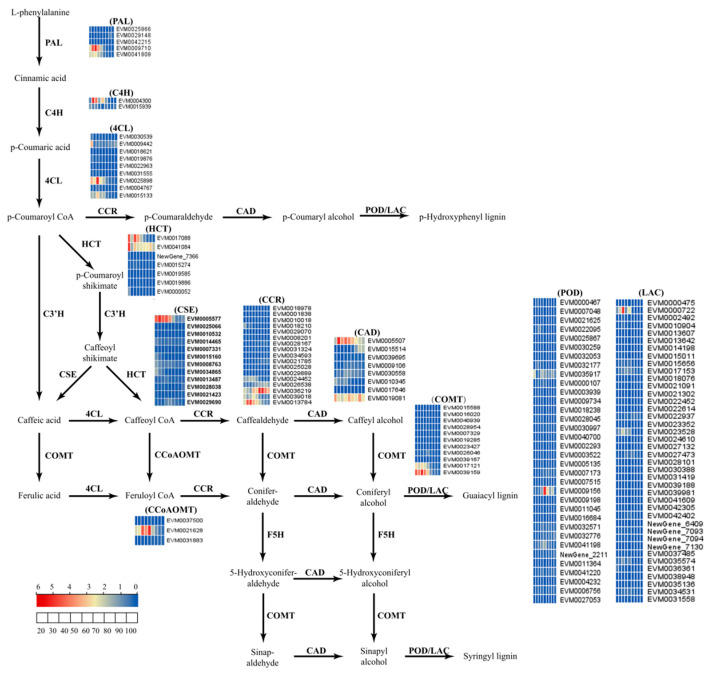
The expression patterns of structural genes involved in the lignin biosynthesis pathway in the ‘Xinyu’ peel during fruit development. The heatmaps indicate the expression levels of the structural genes at different periods of fruit growth and development of the ‘Xinyu’ pear. FPKM (fragments per kilobase of transcript per million fragments mapped) is an expression unit that represents the expression level of genes in a transcriptome. The percentile value of FPKM/relative content values of structural genes ranging from low to high is represented by blue to red in the bottom left corner.

**Figure 7 ijms-25-07481-f007:**
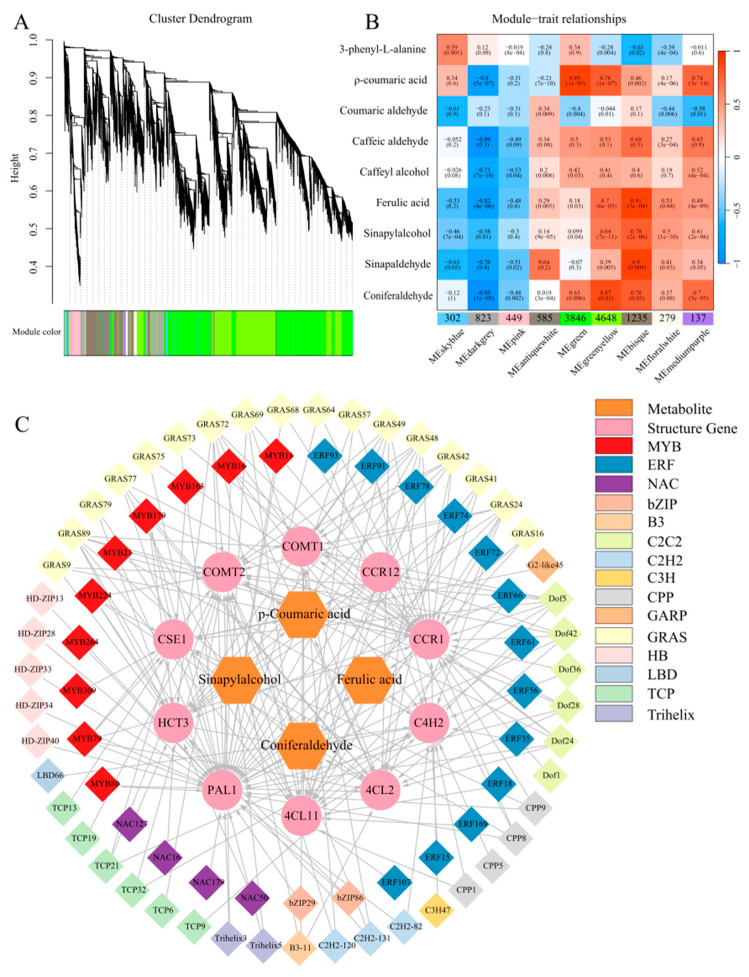
WGCNA of the DEGs identified at the different development stages of the ‘Xinyu’ pear. (**A**) Hierarchical clustering tree showing nine co-expressed gene modules. A total of 12,304 DEGs are clustered into branches, and each module is represented by the main tree branch. The lower panel displays the module in the specified color. (**B**) Module–trait correlations and the corresponding *p*-values are in parentheses. The left panel shows nine modules. The color scale on the right indicates the correlation of module characteristics from −1 (blue) to 1 (red). The panel labeled ‘lignin’ represents the biosynthetic properties of lignin. The other panels represent changes in gene expression levels. (**C**) The gene network of the hub genes in the green–yellow module, which were positively correlated with the lignin content. The correlation network diagram is divided into four layers, the two outermost genes are transcription factors, the third layer is lignin synthesis genes, and the most central layer is metabolites of lignin.

**Figure 8 ijms-25-07481-f008:**
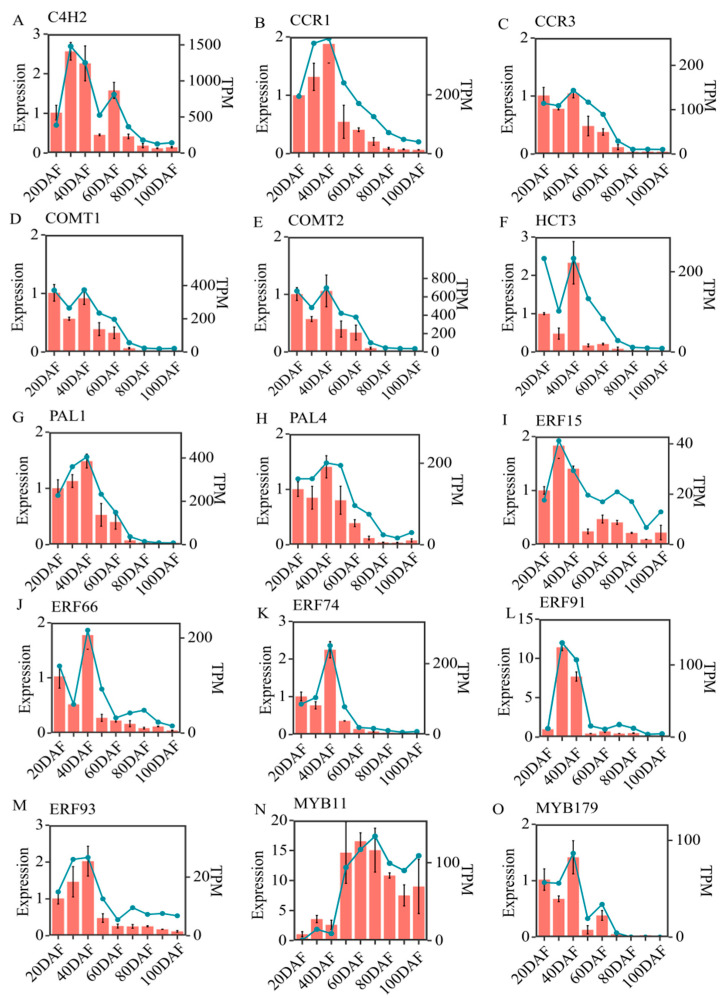
RT-qPCR analysis of eight differentially expressed structural genes and seven transcription factor genes related to lignin biosynthesis in ‘Xinyu’ pear peel during the different development stages. The bar plots illustrate the outcomes of the RT-qPCR analysis. Bars represent means ± SD (n = 3). The line graph represents the mean value of TPM for each gene from the transcriptomic data.

**Table 1 ijms-25-07481-t001:** The contents of lignin-related metabolites (μg/g) in the ‘Xinyu’ pear peel at the different stages of fruit development.

Substance	Different Stages of Fruit Development								
	20DAF	30DAF	40DAF	50DAF	60DAF	70DAF	80DAF	90DAF	100DAF
Coumaric aldehyde	0.09 ± 0.01	0.13 ± 0.01	0.11 ± 0.01	0.08 ± 0.01	0.13 ± 0.01	0.12 ± 0.01	0.12 ± 0.01	0.10 ± 0.00	0.09 ± 0.02
Ferulic acid	0.03 ± 0.01	0.05 ± 0.00	0.05 ± 0.00	0.05 ± 0.00	0.05 ± 0.00	0.04 ± 0.00	0.04 ± 0.00	0.03 ± 0.01	0.02 ± 0.00
Sinapyl alcohol	0.01 ± 0.00	0.02 ± 0.00	0.05 ± 0.01	0.03 ± 0.00	0.02 ± 0.00	0.02 ± 0.00	0.01 ± 0.00	0.01 ± 0.00	0.01 ± 0.00
Sinapaldehyde	0.49 ± 0.16	0.98 ± 0.02	1.18 ± 0.17	1.16 ± 0.13	1.40 ± 0.02	1.05 ± 0.11	0.675 ± 0.09	0.41 ± 0.06	0.26 ± 0.06
L-phenylalanine	143.33 ± 3.40	53.40 ± 1.31	21.40 ± 2.16	39.30 ± 2.57	43.27 ± 1.54	89.83 ± 2.31	80.77 ± 14.47	96.27 ± 19.07	51.77 ± 12.19
p-coumaric acid	1.20 ± 0.03	0.77 ± 0.07	0.78 ± 0.09	0.70 ± 0.05	0.58 ± 0.026	0.44 ± 0.04	0.35 ± 0.04	0.22 ± 0.03	0.16 ± 0.05
Caffeic aldehyde	0.05 ± 0.01	0.04 ± 0.00	0.04 ± 0.01	0.04 ± 0.01	0.04 ± 0.01	0.04 ± 0.02	0.02 ± 0.00	0.02 ± 0.01	0.01 ± 0.00
Caffeyl alcohol	0.05 ± 0.00	0.04 ± 0.00	0.05 ± 0.01	0.04 ± 0.00	0.04 ± 0.00	0.05 ± 0.00	0.04 ± 0.00	0.04 ± 0.01	0.03 ± 0.01
Conifer aldehyde	0.98 ± 0.19	1.04 ± 0.10	1.12 ± 0.15	1.00 ± 0.07	0.83 ± 0.11	0.69 ± 0.15	0.53 ± 0.01	0.38 ± 0.09	0.25 ± 0.05

Note: The value in this table is the area under the peak that is meant to indicate abundance. Data are presented as the mean ± standard deviation (n = 3).

## Data Availability

The raw sequencing data of RNA-seq were deposited in the National Genomics Data Center (2024) database (https://www.cncb.ac.cn) with BioProject number “PRJCA023750”.

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
