# Peer review of "A Combined Metabolome and Transcriptome Reveals the Lignin Metabolic Pathway during the Developmental Stages of Peel Coloration in the ‘Xinyu’ Pear"

_ijms, 2024, doi:10.3390/ijms25137481_

Round 1

Reviewer 1 Report

Comments and Suggestions for Authors

Dear Authors, I have reviewed the manuscript and I will provide my comments below.

The topic of the manuscript is the exploration of the mechanism of brown husk formation in Chinese sand pear (Pyrus pyrifolia), which has been studied in a species called Xinyu. The biosynthesis and accumulation of lignin at nine developmental stages were investigated by metabolomic and transcriptomic methods. The topic is very interesting and I think we can count on a large number of reader visits because of the high proportion of research on food crops, including fruits.

However, I do have some suggestions, which I make below:

The Abstract is very long. It would be advisable to shorten it, given that the MDPI sets a maximum of 200 words. I suggest reorganising, condensing and rethinking the chapter. The aim, material, method, results and conclusions should be presented here.

Key words: anything that is not a proper noun or abbreviation should be written in lower case. 

Tables and figures in the in-text references should not be in bold. 

The Introduction chapter is fine. 

References are marked as being omitted in the text.

The Discussion chapter should follow the structure of the Introduction chapter. In addition, please rewrite this chapter with the deeper thoughts of the chapter. 

Reviewer 2 Report

Comments and Suggestions for Authors

The reviewed manuscript, titled “Combined Metabolome and Transcriptome Reveals the Lignin Metabolic Pathway during Developmental Stages of Peel Coloration in ‘Xinyu’ Pear” by Jiang et al. is a basic study that is focused on the identification of the production of lignin and its accumulation within a pear species.  This involves a comprehensive time-course where basic metabolic profiling was performed and a number of transcriptomic methods were employed to verify and validate this process.  Overall I find this work to be a bit disjointed in the science and the reporting.  There is a lack of conciseness that is present in most of the sections, and a lack of pertinent details seen in sections that are needed to understand the context.  I believe that this work is fundamentally sound, but could benefit from a number of revisions and clarifications that I have outlined below.  One strength of this work – and one that this reviewer would like to commend and acknowledge – are the transcriptomics (RNA-Seq, analysis, and qRT-PCR validation studies).  Those sections were very well done.  I believe that this work would be an appropriate addition to the literature upon revision.

Major comments:

The abstract needs to be revised for simplicity, conciseness, and clarity.  It is not typical or appropriate to list every finding observed in the abstract.  As a result, this abstract does not read well or conform to established norms.  Aim to provide an overview and the key findings in this section.  It is not necessary to list everything and every possible TF and targets.

Line 122: “Different letters indicate a significant difference at p < 0.05.” please explain more thoroughly.  I interpret the meaning that the bars with the same letters do not have a significant difference from each other, but any two bars with a different letter have a significant difference that is below the 0.05 threshold.  Please clarify this in the legend if it is true (or not).  Additionally, I don’t know how to interpret the bar at 60 DAF that has both ab above it. 

Line 125: “As shown in result, complete pores could be seen on the surface of the pear peel, and the wax film on the peel was relatively smooth with small gaps at 20DAF (Figure 3). The peel began to form fruit pots, which were the cork filling cells burst through the surrounding epidermis and gradually fill the cavity, but the fruit surface still was smooth. The fruit spots gradually increased and unevenly formed a rough cuticle film and corkification. It was difficult to observe pore at 50DAF.”  Add arrows to the EM figures to highlight the specific features you are discussing at each stage of development.

Line 155: “Chromatographic peak area of metabolites in ’Xinyu’ peel at different stages of fruit growth and development”.  This is unclear.  What exactly are you conveying?  The text reads that the peak area is shifting, is that correct?  An area peak from 143 (20DAF) to 21.4 (40DAF) for phenylalanine would seem to be a drastic shift.  Is this meant to indicate that the specific peaks from your MS/MS analysis have shifted that much?  Or is this an area under the peak that is meant to indicate abundance?  This is quite unclear.

Line 158: what is the difference between table 1 and this figure?  Why is the scale from 2 to -2?  Is there a normalization step that occurred to get his to the range presented?  This is unclear to this reviewer.

Discussion is fine, but overly long.  It is not necessary to reiterate every result a second time.  I would recommend revising for clarity and conciseness.  I would cut this section in half.  Methods are fine and appropriate.

Minor comments:

Line 18: “Excepted for L-phenylalanine, these metabolites obvious showed a developmental specificity…” poor grammar and tense agreement.  Unclear meaning.

Line 41: “The color of fruit peel is one of important agronomic traits and commodity qualities of pear” poor grammar and meaning diminished.

Line 180: “The PCA and cluster dendrogram of the transcriptomic data also supported the classification of gene expression patterns into these two groups (Figure 2B;2C).”  Figure two lacks panels B and C.

Comments on the Quality of English Language

Language needs work.  It is largely fine, but there are consistent grammatical and tense inconsistencies that diminish the reading and need to be corrected.  I believe that the authors could accomplish this themselves with a careful review and copy editing.

Reviewer 3 Report

Comments and Suggestions for Authors

The work is a facts paper-  how is the lignin formed in pear peel

Thus the work pertains to a very specfic focussed research area.  There was no hypothesis  for the work

The pear peel background could be better introduced -  most of the information is set down only in the discussion

why would such a focussed study be important?  as shown in paper  the pathways for lignin synthesis  is quite well documented already  

Scientific problems nclude:

no discussion on any replication  data -  limited  stat analysis       no idea how robust the data are 

what are the environmental cues to start peel ligninification?

how does this information relate to the browning coloration? 

the images are poor  (mainly size  and incompleteness) and there are no details in legends  

the supplemental  has no corresponding wording to show what the data are and how obtained      inadequate based on other papers  

Comments on the Quality of English Language

some sentences are wrongly composed-  facts are proveded but these are not integrated into a proper sentence        some are noted with sticky notes

there are several spelling and spacing problems  

Round 2

Reviewer 1 Report

Comments and Suggestions for Authors

The parts of the manuscript that I suggested have been modified - I recommend the manuscript for publication. 

Author Response

Thanks for your cooperation and valuable comments!

Reviewer 2 Report

Comments and Suggestions for Authors

I appreciate the effort that the authors made to revise the manuscript.  I believe that they have successfully addressed my previous concerns in an appropriate and adequate fashion.

Author Response

(The authors gave the same response as above.)

Reviewer 3 Report

Comments and Suggestions for Authors

Please again use a  professional editor -  there are still writing and format problems   

the supplementary work  lacks any corresponding methods   or comments 

paper is still factual-  conclusion reads like a summary of results not a conclusion   how does this work become useful

we already  knew about how lignin is formed and some of the basics about regulation for the pathways for lignin synthesis

one aspect that would be useful would be more of a scientific discussion of cork  and lignin     is it only the cork cells that become lignified  in the browned peel  or are other cell walls too 

there is sometimes vocabulary use that is very specialized for this crop 

the figure qualities remain poor    -  they must be magnified to read words but then the image is very blurred   so problems remain  

science

predictable due to lots of prior work

no hypothesis      minimal replication 

discussion remains factual   ( there is comparison with other systems) 

one of the few papers though that attemps to relate product with synthesis 

Comments on the Quality of English Language

Author Response

Thanks for your cooperation and valuable comments! We have revised the manuscript according to your comments and re-submitted it. The detail of point-by-point revision was attached below.
